# Should Peak Dose Be Used to Prescribe Spatially Fractionated Radiation Therapy?—A Review of Preclinical Studies

**DOI:** 10.3390/cancers14153625

**Published:** 2022-07-26

**Authors:** Cristian Fernandez-Palomo, Sha Chang, Yolanda Prezado

**Affiliations:** 1Institute of Anatomy, University of Bern, 3012 Bern, Switzerland; cristian.fernandez@unibe.ch; 2Department of Radiation Oncology, University of North Carolina School of Medicine, Chapel Hill, NC 27599-7512, USA; 3Institut Curie, Université PSL, CNRS UMR3347, Inserm U1021, Signalisation Radiobiologie et Cancer, 91400 Orsay, France; yolanda.prezado@curie.fr; 4Université Paris-Saclay, CNRS UMR3347, Inserm U1021, Signalisation Radiobiologie et Cancer, 91400 Orsay, France

**Keywords:** SFRT, MBRT, MRT, valley dose, peak dose, ILS

## Abstract

**Simple Summary:**

This review study analyzed preclinical studies available before 2022 on spatially fractionated radiation therapy (SFRT), a promising cancer therapy with a high therapeutic index. We intend to use the results of preclinical studies to shed light on the correlation between SFRT dosimetry and treatment response in the clinic. In particular, we challenge the use of peak dose when prescribing SFRT.

**Abstract:**

Spatially fractionated radiotherapy (SFRT) is characterized by the coexistence of multiple hot and cold dose subregions throughout the treatment volume. In preclinical studies using single-fraction treatment, SFRT can achieve a significantly higher therapeutic index than conventional radiotherapy (RT). Published clinical studies of SFRT followed by RT have reported promising results for bulky tumors. Several clinical trials are currently underway to further explore the clinical benefits of SFRT. However, we lack the important understanding of the correlation between dosimetric parameters and treatment response that we have in RT. In this work, we reviewed and analyzed this important correlation from previous preclinical SFRT studies. We reviewed studies prior to 2022 that treated animal-bearing tumors with minibeam radiotherapy (MBRT) or microbeam radiotherapy (MRT). Eighteen studies met our selection criteria. Increased lifespan (ILS) relative to control was used as the treatment response. The preclinical SFRT dosimetric parameters analyzed were peak dose, valley dose, average dose, beam width, and beam spacing. We found that valley dose was the dosimetric parameter with the strongest correlation with ILS (*p*-value < 0.01). For studies using MRT, average dose and peak dose were also significantly correlated with ILS (*p*-value < 0.05). This first comprehensive review of preclinical SFRT studies shows that the valley dose (rather than the peak dose) correlates best with treatment outcome (ILS).

## 1. Introduction

Spatially fractionated radiation therapy (SFRT) is an unconventional radiotherapy (RT) that has been used clinically since the 1970s. In parallel, SFRT has also been the subject of preclinical research for decades [1,2,3,4,5,6,7]. In recent years, SFRT has attracted renewed and enthusiastic interest in radiation oncology [5,7,8,9,10,11,12,13,14]. Compelling data from preclinical research and some, but still limited, clinical studies have indicated that SFRT may lead to a very high therapeutic ratio compared with conventional RT [5,6,7]. Instead of the seamless dose distribution of conventional RT, SFRT has its signature dose distribution, which is characterized by a juxtaposition of many small dose hot spots and cold spots in space in an oscillating pattern [5].

As shown in Figure 1, the geometric dimensions of the multiple beams in preclinical SFRT are very small. Microbeams are the smallest (with a beam width of 100 µm or less) [3,15], whereas minibeams are the largest (with a beam width between 100 µm and 1 mm) [4,16,17]. Numerous preclinical SFRT studies have shown that normal tissue tolerates irradiation exceptionally well, even when the peak dose is 100 Gy or more in a single treatment [17,18,19,20]. The sparing of normal tissue in conjunction with the tumor ablation of preclinical SFRT has been shown to significantly prolong animal survival [6,15,21,22,23,24]. For clinical application, SFRT is delivered in larger geometric dimensions (peak width in the order of 1 cm), originally in the form of GRID therapy [8], as shown in Figure 1, and later in the form of lattice therapy for 3D fractionation (not shown) [9]. In clinical SFRT, a single SFRT delivery is often followed by a course of conventional chemoradiation therapy. In patients with bulky tumors, both effective palliation and tumor control have been reported [2,11,12,25].

SFRT remains largely unused despite its success in the clinic and its proven technical feasibility in virtually every RT unit in the world. Several obstacles stand in the way of the potentially widespread clinical use of SFRT. These include a lack of understanding of its mechanisms of action and the correlation between clinical and preclinical SFRT dosimetry and treatment response. We have already obtained this important knowledge with regard to conventional RT and use it routinely in RT treatment planning (set dose tolerances for normal tissue based on the extensive clinical data summarized in QAUNTEC [26] and ensuring that the high dose covers the entire tumor volume, knowing that the minimum tumor dose dominates tumor control [27]). However, we do not have such an understanding of the correlation between dosimetry and treatment response to guide and optimize the clinical application of SFRT. Our understanding to date strongly suggests that the dominant mechanism of action of clinical SFRT may be very different from that of conventional RT [14]. Cytotoxic radiation cell killing, which is dominant in seamless radiation therapy, may overpower the radiation-induced secondary effects such as the cellular-level bystander effect, tissue-level tumor microenvironment modulation, and the systemic immune system modulation that may lead to the abscopal effect. In SFRT, however, these secondary effects may play much more significant roles compared to conventional RT. A better understanding of SFRT is critically needed for its potentially broad clinical application. The fact that the dosimetry of SFRT is much more complex than that of conventional RT, further complicates matters. In a recent preclinical study by Rivera et al. that examined the correlation between SFRT dosimetry and treatment response, seven parameters were used to describe SFRT dosimetry [17]. Radiation was delivered to a rat sarcoma tumor model using a wide range of spatial fractionation scales (1 cm to 300 microns) at the same integral tumor dose. The above study showed that of the seven dosimetric parameters examined, the valley dose had the strongest correlation and the peak dose had the weakest correlation with tumor control and survival [17]. This finding from a single study challenges the appropriateness of using the peak dose for prescribing SFRT. This review is motivated by the question of whether a similar correlation between dose and treatment response can be found in all relevant preclinical studies on SFRT.

To date, most SFRT regimes have been prescribed according to the peak dose, although we do not know whether the peak dose is the dosimetric parameter of SFRT that is closely associated with tumor response or normal tissue toxicity. If the peak dose is indeed poorly correlated with treatment response, it means that the same treatment regimen based on the peak dose will result in a range of different treatment outcomes. In this case, another dosimetric parameter that has a strong correlation with treatment response should be used to prescribe SFRT. It is a major challenge to study the correlation between SFRT dosimetry and treatment response in clinical practice because SFRT is usually followed by a full course of conventional RT [2,11,28]. This makes it difficult to determine the specific impact of SFRT dosimetry on clinical treatment response. Nevertheless, understanding the correlation between SFRT dosimetry and treatment response is critical for optimizing SFRT and designing clinical trials. The simplicity of preclinical SFRT research, using a single fraction of SFRT, facilitates the study of the correlation between dosimetry and treatment response. Although the results of preclinical SFRT are not directly translatable to the clinic, we hypothesize that the correlation between SFRT dosimetry and treatment response identified in preclinical studies may shed important light on understanding the correlation in clinical applications.

It is important to point out that the peak dose captures only one aspect of the complex SFRT dosimetry, which is characterized by a high degree of spatial fractionation of the radiation. In addition to the peak dose, the valley dose and the peak and valley width are also among the basic dosimetric parameters of SFRT. Whether the peak dose is the dosimetric parameter most closely associated with a given treatment response is a crucial question that we need to answer for a robust and optimized SFRT regimen. In this review article, we evaluated whether the findings of a single animal study [17] are representative of preclinical SFRT (MRT and MBRT) in general. To the best of our knowledge, we used the data available before 2022.

## 2. Materials and Methods

### 2.1. Background and Objective

The aim of this review study was to reveal the correlation between dosimetric parameters of SFRT and treatment response in previous preclinical SFRT studies performed by different research groups using different animal models and forms of SFRT delivery before 2022. Most of the studies included in this review were designed for other study objectives, had different endpoints, and reported different (and often incomplete) dosimetric parameters than what were desired for this review. To minimize this limitation, we homogenized the data by using increased life span (ILS) as the endpoint of our evaluation. ILS is defined in Equation (1), where MST refers to median survival time.
(1)$$\mathrm{ILS}=\frac{\left({\mathrm{MST}}_{\mathrm{treated}\;}-{\mathrm{MST}}_{\mathrm{controls}}\right)}{\mathrm{MST}\;\mathrm{controls}}\times 100$$

With ILS, each study can be normalized to its own control group under the same conditions (animal model, tumor model, etc.). In this way, we can minimize the impact of the many differences between studies (in terms of animal, tumor model, type of radiation used, etc.) when examining the correlation between dosimetric parameters and the treatment response.

### 2.2. Eligibility Criteria

We reviewed previous studies that used MRT or MRBT on tumor-bearing small animals before 2022. The full list of studies that we reviewed for this work can be found in the Appendix A. The inclusion criteria are listed below:-Studies used a single array of MRT or MBRT;-Studies reported survival data as either mean or median survival;-Studies reported appropriate dosimetric information and radiation settings (peak dose, valley dose, peak width, and spacing);-Peak doses of less than 800 Gy were used in the studies (this criterion was intended to avoid confounding factors related to toxic effects of radiation exposure that are unlikely to be relevant to the clinical application of MRT or MBRT);-MRT or MBRT were not administered in combination with other therapies, such as nanoparticles or gene therapy.

Publications reporting only tumor ablation without a control group were not included in this review because we cannot calculate ILS (Data curation_MRT, Data_curation_MBRT). Only one publication containing the desired dosimetric parameters from the Brookhaven National Laboratory (BNL) was found. We excluded this publication because of concerns about the accuracy of the dosimetry in the dated publication (there have been no new Monte Carlo calculations or dosimetry measurements to validate the MRT work at BNL in the last 16 years).

### 2.3. Search Strategy

As a starting point, we reprocessed the MRT data collected in the recently published scoping review by Fernandez-Palomo et al. [15]. We then performed a comprehensive literature search to identify MBRT studies and any new MRT publications. We also included a few unpublished studies that are soon-to-be published to increase the statistical power. These studies are clearly identified in Table 1 and Table 2.

### 2.4. Data Curation

Table 1 and Table 2 show the lists of MRT and MBRT studies, respectively. As part of the curation process, relevant dosimetric parameters that were not directly reported in the papers were derived either from other parameters in the study (indicated in red in the tables) or from related studies that used the same SFRT machine (indicated in green in the tables). In MRT experiments, the peak and valley doses were reported at the site of entry and/or in depth.

### 2.5. Data Analysis

RStudio was used to plot the distribution of the data used in this review study in terms of radiation facilities for ILS, valley dose, peak dose, PVDR, and average dose (Figure 2). These boxplots are a graphical representation of the data to facilitate the interpretation of the correlations that follow.

A principal component analysis (PCA) was performed to examine the data and see if an influential variable could be identified. We chose PCA because our data set, while not large, is high-dimensional. The high dimensionality comes from the many dosimetric and geometric SFRT parameters, while the observations are few and come from different animal models.

PCA was performed in RStudio using the “missMDA” package. Our dataset contained missing values (e.g., some papers did not report the valley dose). These missing values were iterated using 2 algorithms: “estim_ncpPCA” to estimate the number of dimensions, and “imputePCA” to calculate the missing values based on the number of dimensions from the first algorithm. Once the missing values were imputed, the PCA was calculated and drawn using the algorithm “PCA”.

The results of the PCA are presented visually as a plot of the first two components (or dimensions) with the greatest variability in the data set.

One-tailed Pearson correlation analyzes were performed using GraphPad Prism 9 (GraphPad Software, version 9.4.0, CA, USA). Pearson r, r^2^, and *p* values were calculated. First, all data that met the eligibility criteria were analyzed, regardless of the irradiation technique. In a second step, the data were divided depending on the technique used (MRT or MBRT).

We used the Pearson correlation to investigate whether there is a linear relationship between the ILS and any of the physical parameters of preclinical SFRT. The strength of this linear relationship is reflected in the correlation coefficient r (as a value between −1 and +1). The proportion of the variance of a given preclinical SFRT parameter that predicts the variance of the ILS is represented as r^2^. Because we use physical preclinical SFRT parameters to statistically infer their influence on ILS, we must assume that there could be biological parameters (variables) not considered in this study that could also influence ILS (e.g., immune cell influx, degree of vascular permeability, etc.). In such cases, where not much variance in the dependent variable is predictable, Cohen [29] recommends using the operational definitions of “small”, “medium”, or “large” in terms of the coefficient r. Thus, we used the following definitions in the analysis of our data:Small correlation: r ≤ 0.1Medium correlation: 0.1 > r < 0.5Large/strong correlation: r ≥ 0.5

## 3. Results

The principal component analysis in Figure 3 shows that valley dose, average dose, peak dose, and PVDR have a positive association with the first component/dimension (Dim1, which accounts for 52.24% of the variance in the data). This suggests that Dim1 may represent “dose deposition.” PCA also revealed that ILS and valley dose had a positive association with the second component/dimension (Dim2, accounting for 17.85% of the variance in the data). This suggests that Dim2 may represent the treatment outcome. In summary, the PCA shows that valley dose is the most influential variable in the data.

Table 3 shows the compilation of the regression coefficients of ILS versus each of the dosimetry and geometrical parameters considered. The data was analyzed as a whole “ALL”, and also divided into “MRT” and “MBRT”.

No strong correlation was found between ILS and any of the dosimetric parameters when all data were included in the analysis (ALL), regardless of the preclinical SFRT mode used. However, strong correlations appeared when MRT and MBRT were analyzed separately.

As for MRT, valley dose was the parameter with the strongest correlation with ILS (r = 0.822 and *p* = 0.0018) where 68% of the variance of valley dose (r^2^ = 0.675) can predict/influence the variance of ILS. Average dose also showed a strong correlation with ILS (r = 0.683 and *p* = 0.0147), while peak dose had a medium correlation with ILS (r = 0.485 and *p* = 0.0333). We found that some of the dosimetric parameters used in the preclinical SFRT studies were closely correlated (not independent variables), and we analyzed the correlations (see graph 2 in the Appendix A). For instance, we found that average dose had a significant correlation with valley dose (r = 0.779, r^2^ = 0.608, *p* = 0.0039). There was no strong correlation between ILS and other dosimetric parameters, especially PVDR. Our findings challenge the long-held assumption that PVDR plays an important role in the treatment response of MRT [21], which has been widely shared among MRT researchers.

Concerning MBRT, the only parameter that showed a strong correlation with ILS was valley dose.

To reduce the large heterogeneity of the studies, we analyzed a subset of the studies that treated brain tumors only, the largest subset in this review. The results are shown in Table 4.

For the “MRT brain tumor only” subset, the valley dose shows the next strongest correlation with ILS, followed by the average dose (*p* = 0.0021 and *p* = 0.0205, respectively). The spacing shows a strong correlation, albeit negative, with ILS in MRT (*p* = 0.0005).

In the “MBRT brain tumor only” subset, valley dose and peak width/valley width showed a strong correlation with ILS, although it was not significant, which could be related to the fact that proton and synchrotron radiation as well as conventional X-rays were used.

When we analyze MBRT independently (Table 5), some strong correlations are observed in the proton MBRT studies (all brain tumor treatments).

In proton MBRT, valley dose strongly correlates with ILS (*p* = 0.0030), while peak dose and PVDR had a strong negative correlation with ILS (*p* = 0.025 and *p* = 0.002, respectively). As shown in Table 5, no correlations were found in the photon MBRT group, which is probably related to more heterogeneous data (different tumor models, different facilities, etc.).

## 4. Discussion

Decades of preclinical studies have indicated that the microbeam and minibeam forms of SFRT have a significantly higher therapeutic index compared with conventional RT. Although strong clinical data supported by randomized trials is currently absent, clinical studies published so far using GRID and lattice therapy have indicated the potential of a higher therapeutic index for clinical cases challenging for conventional RT, especially when it is followed by conventional RT [11,30]. In recent years, SFRT has attracted renewed interest in the RT community, and a number of clinical trials are planned or underway [14]. However, our understanding of clinical and preclinical SFRT in terms of its mechanism of action and the correlation between dosimetry and treatment outcome remains very limited. Without this important knowledge, we cannot meaningfully optimize the clinical SFRT regimen. Compared to the dosimetry of conventional RT, the dosimetry of clinical SFRT is significantly more complex. Clinical and preclinical SFRT-specific dosimetric parameters include peak dose and valley dose, peak width and valley width, from which other parameters can be derived, such as peak to valley dose ratio and the percentage of directly irradiated volume. Currently, we do not know which of the dosimetric parameters of preclinical SFRT are clinically important and should be optimized for best tumor control and lowest toxicity. We hypothesize that the dosimetric parameters that capture the essence of preclinical SFRT dosimetry (i.e., the coexistence of hot and cold dose subregions) will have a closer correlation with treatment response than the dosimetric parameters that do not capture this essence. Furthermore, we hypothesize that not all preclinical SFRT-specific dosimetric parameters have the same correlation with treatment response, and thus have the same clinical significance. This is demonstrated by our review, which shows that PVDR, a preclinical SFRT-specific dosimetric parameter that characterizes a major aspect of radiation spatial fractionation, has no significant correlation with ILS. Further careful studies are needed to unravel the correlation between SFRT dosimetry and treatment response.

In the limited clinical trials conducted to date with GRID RT [5], the lack of a control arm, the presence of a heterogeneous patient population, the uncontrolled combination with conventional RT, and the prescription method represent some important limitations of the studies [5]. In particular, dose prescription is usually at the depth of the maximum dose deposition in the peak region. Consequently, the same GRID geometry and dose prescription can result in very different doses to both the target and organs-at-risk, depending on the specific anatomical context. Treatment prescription used in lattice therapy is a significant improvement compared to GRID therapy, as the 3D dose vertices (peaks) inside the treatment targets are used for treatment prescription. Recently, there are encouraging attempts to establish guidelines for dose prescription and reporting in clinical SFRT [8]. However, these guidelines do not deviate from the usual dose prescription method that is based on the peak dose; implying that the peak dose is a dosimetric parameter that is closely related to treatment response. To our knowledge, this is not supported by any comprehensive or systematic analysis of clinical or preclinical data. Therefore, the main objective of this critical review was to shed light on this important issue by examining a broad range of preclinical SFRT studies.

Our analysis shows that valley dose is the variable with the most influence in our dataset (Figure 3) and with the strongest correlation with treatment response (studied as ILS) in both MRT and MBRT (Table 3 and Table 4). In contrast, peak dose had a medium correlation with ILS, but only in MRT studies and not at all in MBRT studies. These findings from the critical review of 17 preclinical SFRT (MRT or MBRT) studies are consistent with the results of the study by Rivera et al. [17] on the correlation of preclinical SFRT dosimetry and treatment response, which showed that the valley dose had one of the highest correlations and the peak dose had one of the lowest correlations with animal survival and body-weight loss in a rat sarcoma tumor model. Our findings from this preclinical SFRT review study challenge the current practice of clinical SFRT treatment prescription. We recognize that the peak dose may be an important clinical and preclinical SFRT dosimetric parameter for cytotoxic cell killing and for triggering important secondary radiobiological processes, such as vascular permeability or immune cell infiltration, all of which merit further investigation.

We recognize that there are important differences between preclinical SFRT studies and clinical SFRT application. The size of the peak and valley width (in cm) is significantly larger in clinical SFRT than in preclinical studies (10 s to 100 s microns); the peak to valley dose ratio is significantly lower (~5) in clinical SFRT than in preclinical studies (>10 in MBRT and 20–50 in MRT). More importantly, clinical SFRT is often followed by a course of conventional RT, whereas animal studies often have a single SFRT regime. Additionally, the fact that cancer patients may respond very differently to the same treatment than animal models is also a well-known limitation of preclinical studies. However, our results on the correlations between ILS and the different dosimetric parameters obtained from the preclinical data raise the question of whether the peak dose should be used to prescribe clinical SFRT. If the peak dose does not correlate closely with the outcome of clinical SFRT (i.e., the same peak dose may result in different treatment responses), we should move to a better method of prescribing clinical SFRT.

This review study has some limitations. All studies included in this review, except one, were not designed to examine the correlation between dosimetry and treatment response. Therefore, the available data needed for this review are sparse and heterogeneous. Dose uncertainties in some of the studies due to the complexity of MRT and MBRT dosimetry could also have an impact. However, we estimate that this uncertainty would not change the main conclusion of our study.

Future experiments designed to investigate the correlation between clinical and preclinical SFRT dosimetry and treatment response are needed to further confirm our results. Studies that can separately examine the correlation of dosimetry with tumor control and with treatment toxicity under preclinical SFRT conditions that resemble clinical use are highly desirable.

## 5. Conclusions

This comprehensive review of preclinical SFRT work performed prior to 2022 shows that the valley dose, rather than the peak dose, is the dosimetric parameter that correlates best with treatment outcome (increased life span) in both MRT and MBRT studies.

## Figures and Tables

**Figure 1 cancers-14-03625-f001:**
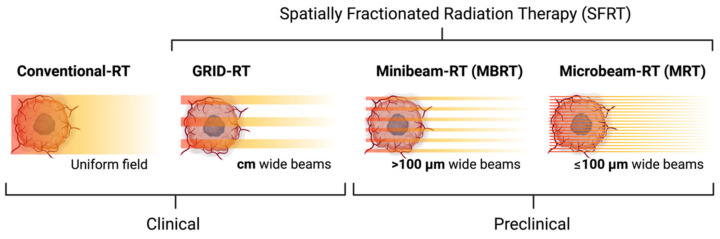
Illustration of radiation spatial distribution of SFRT. Starting from conventional RT where radiation is seamless, SFRT radiation is spatially fractionated in increasing smaller scales—from clinical GRID-RT (and lattice therapy not shown) to preclinical MBRT to MRT.

**Figure 2 cancers-14-03625-f002:**
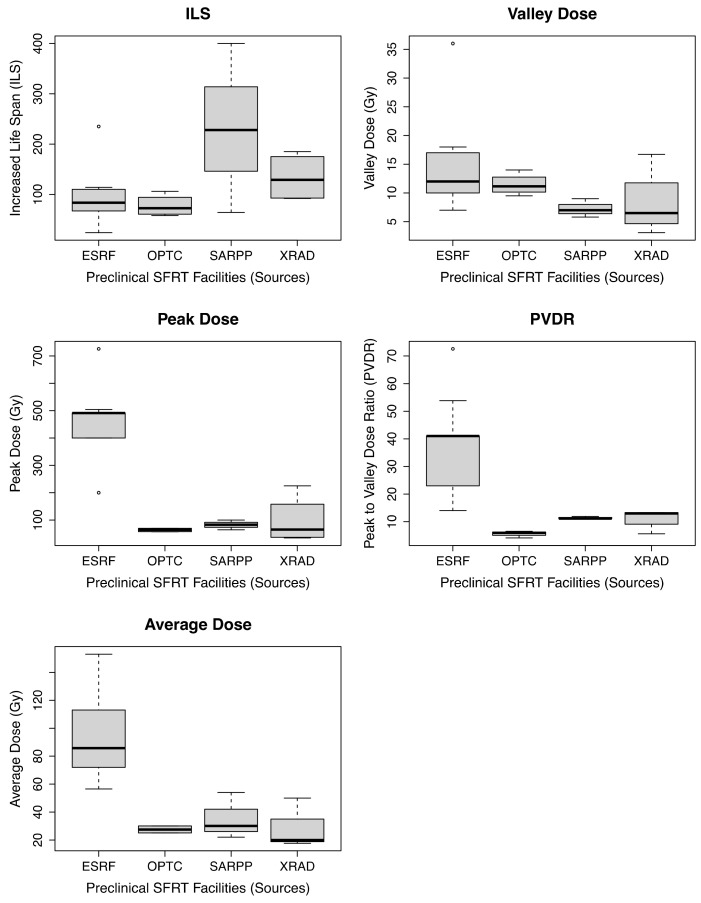
Comparative distribution of the data among radiation facilities. No statistical analysis was performed. RStudio was used to create these plots.

**Figure 3 cancers-14-03625-f003:**
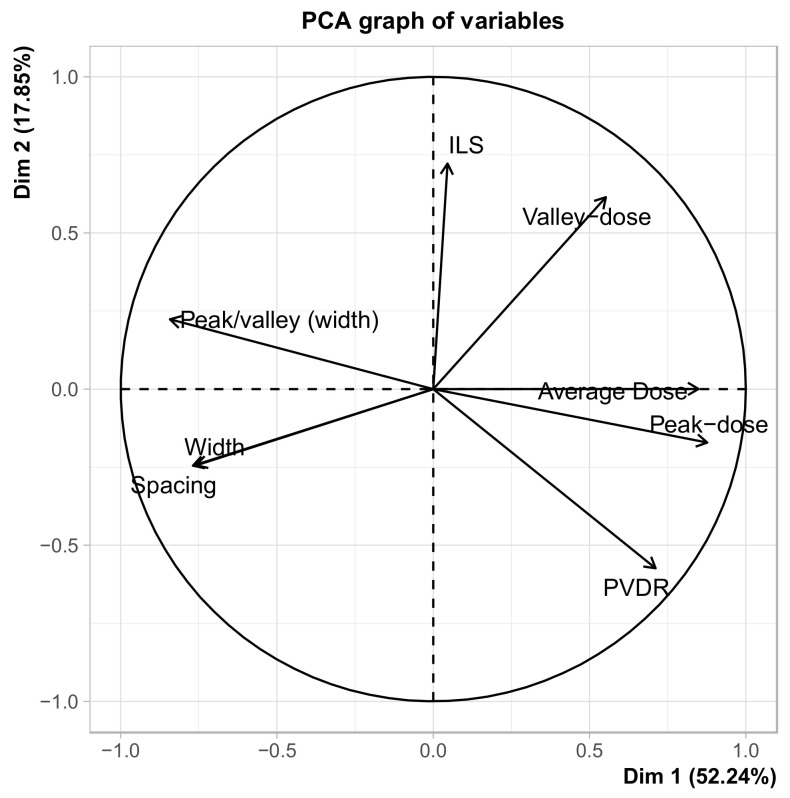
Principal component analysis (PCA) of all variables included in this preclinical SFRT study. Dimension or Component 1 (Dim1) accounts for 52.24% of the variance in the data and could represent dose deposition (dosimetry) as valley dose, average dose, peak dose, and PVDR have a positive effect on it. Dimension or Component 2 (Dim2) accounts for 17.85% of the variance in the data and could represent treatment outcome as ILS and valley dose have a positive influence on it.

**Table 1 cancers-14-03625-t001:** MRT studies and their dosimetry parameters included in this evaluation.

Reference	Animal	Tumor Type	Width (µm)	Spacing (µm)	Peak Width/Valley Width	Peak-Dose (Gy) (at Depth)		Valley-Dose (Gy) (at Depth)		Average Dose (Gy)	PVDR	Increased Life Span (ILS)	Radiation Facility
(A Bouchet et al., 2013)	Rat	9LGS	50	200	0.33	400	(7 mm)	18	(7 mm)	113.5	22.2	^ a ^ 74	ESRF
(Bouchet et al., 2016)	Rat	9LGS	50	200	0.33	200	(7 mm)	9	(7 mm)	56.5	23.0	^ a ^ 24	ESRF
	Rat		50	200	0.33	400	(7 mm)	17	(7 mm)	113.1	23.0	^ a ^ 68	ESRF
(Miura et al., 2006)	Mouse	SCCVII	35	200	0.21	442	entrance					^ a ^ 317	ESRF
	Mouse		35	200	0.21	625	entrance					^ a ^ 383	ESRF
	Mouse		35	200	0.21	884	entrance					^ a ^ 583	ESRF
	Mouse		70	200	0.54	442	entrance					^ a ^ 533	ESRF
(Régnard, Duc, et al., 2008)	Rat	9LGS	25	200	0.14	491	(10 mm)	12	(10 mm)	72.0	41	^ a ^ 100	ESRF
	Rat		25	200	0.14	491	(10 mm)	12	(10 mm)	72.0	41	^ a ^ 110	ESRF
	Rat		25	200	0.14	491	(10 mm)	12	(10 mm)	72.0	41	^ a ^ 93	ESRF
	Rat		25	100	0.33	504	(10 mm)	36	(10 mm)	153.0	14	^ a ^ 235	ESRF
(Régnard, Bräuer-Krisch, et al., 2008)	Rat	9LGS	25	200	0.14	^ b ^ 491	(10 mm)	^ b ^ 12	(10 mm)	72.0	41	^ a ^ 61	ESRF
(Smilowitz et al., 2006)	Rat	9LGS	^c^ 27	211	0.15	625	entrance					^ a ^ 19	ESRF
(Fernandez-Palomo et al., 2020)	Mouse	B16-F10	50	200	0.33	401	entrance	7	(0.3 mm)	105.9	53.8	^ a ^ 67	ESRF
(Eling et al., 2021)	Rat	9LGS	50	400	0.14	726	(7 mm)	10	(7 mm)	99.5	72.6	^ a ^ 114	ESRF

^a^ ILS was calculated to homogenize the treatment response across all different studies. ^b^ Peak and valley doses at this depth were not reported in this study. We extrapolated them from the author’s previous publication (Régnard, Duc, et al., 2008), which employed the same field size, energy spectrum, dose rate, and filters. ^c^ This is the median of the microbeams, which were in the range of 20–39 µm in width. Red colors means values calculated by the authors. Blue color represents values not reported but extrapolated from their previous paper.

**Table 2 cancers-14-03625-t002:** MBRT studies and their dosimetry parameters included in this evaluation.

Reference	Animal	Tumor Type	Width (µm)	Spacing (µm)	Peak Width/Valley Width	Peak-Dose (Gy)	Valley-Dose (Gy)	Average Dose (Gy)	PVDR	Increased Life Span (ILS)	Radiation Facility
Prezado et al., 2019	Rat	RG2	n.a.	n.a.	n.a.	26	21	25	1.2	705	Orsay proton therapy centre
Prezado et al., 2018	Rat	RG2	1100	3200	0.52	70	10.8	25	6.5	63	Orsay proton therapy centre
Lamirault et al., Rad Research	Rat	F98	1100	3200	0.52	70	11.5	30	6.0	106	Orsay proton therapy centre
			1100	3200	0.52	58	9.5	25	6.0	58	Orsay proton therapy centre
Sotiropoulous 2021	Rat	RG2	700	1400	1.00	64	5.8	22	11.1	400	Photon SARPP-Curie
accepted Int. J. Rad. Oncol. Biol. Phys.	Rat	RG2	700	1400	1.00	83	7	30	11.9	64	Photon SARPP-Curie
Prezado jsr 2012	Rat	9L	600	1200	1.00	100	9	54	11.1	228	Photon SARPP-Curie
Rivera et al., 2020	Rat	FSA	310	1200	0.35	91	6.8	20	13.3	93	Photon XRAD-UNC
			310	1200	0.35	225	16.7	50	13.3	185	Photon XRAD-UNC
			2200	4000	1.22	34.5	6.2	17.63	5.6	165	Photon XRAD-UNC
			10,000	20,000	1.00	39	3.1	20	12.6	92	Photon XRAD-UNC

Red color represents Value calculated by the authors.

**Table 3 cancers-14-03625-t003:** Correlation coefficients of ILS versus the several dosimetry and geometrical parameters considered in this study. The group “ALL” includes all preclinical SFRT parameters (variables) selected in this study, which are presented in Table 1 and Table 2.

Pearson r/r^2^ (*p*-Value Summary)	ALL (*n* = 16)	MRT (ESRF) (*n* = 8)	MBRT (*n* = 8)
Width	−0.13/0.017 (ns)	0.224/0.05 (ns)	−0.198/0.039 (ns)
Spacing	−0.152/0.023 (ns)	−0.138/0.019 (ns)	−0.243/0.059 (ns)
Peak width/valley width	0.075/0.006 (ns)	0.336/0.113 (ns)	0.364/0.132 (ns)
Peak-dose	0.127/0.016 (ns)	0.485/0.236 (*)	−0.21/0.044 (ns)
Valley-dose	0.32/0.103 (ns)	**0.822/0.675 (**)**	**0.592/0.35 (*)**
Average Dose	−0.198/0.039 (ns)	**0.683/0.467 (*)**	0.008/0 (ns)
PVDR	−0.347/0.121 (ns)	−0.158/0.025 (ns)	−0.453/0.205 (ns)

ns means non-significant, * means a *p*-value of less than 0.05, ** means a *p*-value of less than 0.01, *n* refers to the number of studies included. Strong correlations (r ≥ 0.5) are marked in bold letters.

**Table 4 cancers-14-03625-t004:** Correlation coefficients for the studies including only brain tumors.

Pearson r/r^2^ (*p*-Value Summary)	MRT Brain Tumors Only *n* = 6	MBRT (Protons + Photons) Brain Tumors Only *n* = 7
Width	−0.401/0.16 (ns)	−0.611/0.373 (ns)
Spacing	−0.898/0.806 (***)	−0.617/0.381 (ns)
Peak width/valley width	0.184/0.034 (ns)	**0.641/0.411 (ns)**
Peak-dose	0.239/0.057 (ns)	−0.661/0.437 (ns)
Valley-dose	**0.877/0.769 (**)**	**0.656/0.430 (ns)**
Average Dose	**0.727/0.529 (*)**	−0.134/0.017 (ns)
PVDR	−0.318/0.101 (ns)	−0.442/0.195 (ns)

ns means non-significant, * means a *p*-value of less than 0.05, ** means a *p*-value of less than 0.01, *** means a *p*-value of less than 0.001, *n* refers to the number of studies included. Strong correlations (r ≥ 0.5) are marked in bold letters.

**Table 5 cancers-14-03625-t005:** Correlation coefficients for the studies including only proton MBRT and photons MBRT.

Pearson r/r^2^ (*p*-Value Summary)	Proton MBRT *n* = 4	Photon MBRT *n* = 4
Width	not enough points	−0.314/0.099 (ns)
Spacing	not enough points	−0.326/0.106 (ns)
Peak width/valley width	not enough points	0.187/0.035 (ns)
Peak-dose	−0.949/0.901 (*)	0.034/0.001 (ns)
Valley-dose	**0.994/0.988 (**)**	0.094/0.009 (ns)
Average Dose	−0.268/0.072 (ns)	0.137/0.019 (ns)
PVDR	−0.996/0.992 (**)	−0.172/0.03 (ns)

ns means non-significant, * means a *p*-value of less than 0.05, ** means a *p*-value of less than 0.01. *n* refers to the number of studies included. Strong correlations (r ≥ 0.5) are marked in bold letters.

## Data Availability

The data presented in this study are available in article and Appendix A.

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
