# Peer review of "Should Peak Dose Be Used to Prescribe Spatially Fractionated Radiation Therapy?—A Review of Preclinical Studies"

_cancers, 2022, doi:10.3390/cancers14153625_

Round 1

Reviewer 1 Report

Thank you for the opportunity to review the paper entitled “Should peak dose be used to prescribe spatially fractionated radiation therapy? – a review of preclinical studies”. This paper reviews several pre-clinical microbeam and minibeam RT experiments and looks for correlations between reported dosimetric parameters and increased life span (ILS). In particular, the authors are interested in determining whether the peak dose associated with spatially fractionated RT (SFRT) is the most important parameter related to treatment effect. This is an important question as the current understanding between dosimetry and treatment response in SFRT is very limited and the paper brings forth many good points.  It is well written and organized with no major typographical or grammatical errors. The authors suggest that valley dose is the most important parameters related to ILS, which agrees with several other publications.

I have two major points I would like the authors to consider for revision in addition to several minor points outlined later:

·       Major point #1: There are at least 3 instances within the manuscript where the authors suggest that SFRT has been shown to be superior to conventional RT in clinical applications:

o   In the abstract, lines 19-21: “Preclinical and clinical studies have shown that SFRT can produce a significantly higher therapeutic index than conventional radiotherapy (RT).”

o   In the introduction, lines 40-42: “Compelling evidence from both preclinical research and clinical studies have shown that SFRT can lead to a very high therapeutic ratio compared with conventional RT

o   In the Discussion, lines 237-238: “Decades of studies have shown that SFRT can have a significantly higher therapeutic index compared with conventional RT.”

To my knowledge, there have not been any clinical patient treatments with microbeam or minibeam RT so these modalities could not be considered advantageous over conventional RT in the clinical setting. The only actual in-human treatments have used larger scale “conventional” SFRT methods like lattice RT or grid therapy and of these I am unaware of any clinical randomized controlled trials comparing SFRT to conventional RT. Indeed, as the authors later summarize in lines 254-257, reference #5 states “However, there are also important limitations of the studies described. In most cases, interpretation of these studies is confounded by (1) lack of a control arm;…"

It may be misleading then to state that SFRT has been shown to be better than conventional RT, as there is no strong data yet to support this. There certainly may be preclinical data and anecdotal evidence in the literature, but I suggest the authors revise the bulleted statements above. Doing so would not detract at all from their message, and has the advantage of striking a more neutral tone since there is no comparative data in the setting of a randomized controlled trial to support the notion SFRT is superior to conventional RT. Please note I am not disputing the idea that SFRT may be superior in some instances, I am only suggesting there is no data which has directly confirmed it in patients.

·       Major point #2: I would like the authors to reconsider, or at minimum clarify, the wording and statistical interpretation of their results with regard to correlation and significance of p values.

o   For example in line 197: “Average dose and peak dose also show significant correlation with ILS…”

o   The Pearson coefficient is an indication of correlation, but the p-value doesn’t really tell you much without additional context related to the strength of that correlation. For example, a highly significant p value doesn’t tell you anything about the strength of the correlation and therefore nothing about its practical importance. In other words, I would argue a significant p value for a low value of correlation is not necessarily clinically relevant, especially for the small sample sizes used in this paper. I suggest the authors reframe the wording of their results and cite what is considered strong/weak/no correlation to aid in their interpretation. Consider Cohen J. Statistical Power Analysis for the Behavioral Sciences. 2nd ed. Hillsdale, NJ: Lawrence Erlbaum Associates; 1988 where a correlation of 0.3 - 0.5 is generally considered weak correlation, 0.5-0.7 is considered moderate correlation, and higher values are considered strong correlation. With regard to MRT and peak dose, this changes the message somewhat because the authors could argue peak dose is only “weakly correlated” despite a significant p value. However, valley dose would be “strongly correlated” with ILS and has a significant p value which further supports the idea that valley dose is most important. Similar arguments could be made for MBRT and valley dose as well.

I would ask the authors to please consider the following minor suggestions as well:

Line 89: “To date, all SFRT regimes have been prescribed according to the peak dose…”

·       Please consider avoiding the term “all”, I would suggest “most” instead. There are lattice SFRT protocols that do not prescribe to the peak dose. See for example, Duriseti, S., et al. (2021). "Spatially fractionated stereotactic body radiation therapy (Lattice) for large tumors." Adv Radiat Oncol 6(3): 100639. There are other lattice SFRT papers which do not prescribe to peak dose as well.

Lines 143-145: “Peak doses of less than 800 Gy were used in the studies. This criterion was intended to avoid confounding factors related to toxic effects of radiation exposure that are unlikely to be relevant to the clinical application of SFRT.”

·       I found the choice of 800 Gy to be interesting, especially since one might think toxic effects of radiation exposure in humans could result with peak doses much less than that. Could the authors comment on why 800 Gy was chosen as the cutoff? In particular, what toxic effects are of concern that may manifest only above 800 Gy? If this was an arbitrary cutoff, I think that would be completely fine actually.

·       At times throughout the manuscript, I was somewhat confused as to what modality was being referred to since clinical grid therapy/lattice, MBRT and MRT are all generally characterized as SFRT here. Is there a way to separate these 3 applications of SFRT so it is more clear to the reader which one is being referred to? As an example, consider the statement "...unlikely to be relevant to the clinical application of SFRT." Do the authors mean clinical application of MRT or MBRT? Grid therapy (which is also SFRT) is already in clinical use.

Lines 159- 160: “We also included a few unpublished studies soon-to-be published to increase statistical power.”

·       Can the authors please define “soon to be published?” Have these articles been accepted for publication but are not in press yet? If not, I don’t think they should be included in the analysis until accepted in a peer reviewed journal.

Table 1: All but one of the “Average Dose (Gy)” values are zero. Is this a mistake? Please clarify.

Line 200: “…(see Figure S2 in the supplementary material).”

·       Supplementary materials are not available for review. Will they be provided?

Lines 248 – 250: “We hypothesize that the dosimetric parameters that capture the essence of SFRT dosimetry (i.e., the coexistence of hot and cold dose sub-regions) will have a closer correlation with treatment response…”

·       I agree with this statement, but it is contradictory to lines 203-204: “Our findings challenge the long-held assumption that PVDR plays an important role in the treatment response of MRT [21], which has been widely shared among MRT researchers.” I would suggest most readers would associate “the coexistence of hot and cold dose sub-regions” with PVDR.

·       PVDR is often considered a measure of heterogeneity, and heterogeneity is often considered to be very important for treatment effect. But your analysis suggests there might not be any correlation between ILS and heterogeneity which is quite interesting and perhaps a bit provocative. Would the authors consider a bit more discussion on this potential disparity? Does this mean that PVDR is not a good metric of heterogeneity or that heterogeneity is just not important for the biology?

Line 257: “In particular, dose prescription is usually at the surface…”

·       I am not aware of any clinical SFRT treatments which prescribe dose at the surface of the patient and this would be very difficult to do as most TPS’s have questionable accuracy right at the skin surface. Can the authors please clarify?

Lines 261-262: “Lattice therapy is a significant improvement compared to GRID therapy, as the 3D dose vertexes (peaks)…”

·       I suggest the more commonly used term "vertices" rather than vertexes.

Author Response

REVIEWER 1:

Thank you for the opportunity to review the paper entitled “Should peak dose be used to prescribe spatially fractionated radiation therapy? – a review of preclinical studies”. This paper reviews several pre-clinical microbeam and minibeam RT experiments and looks for correlations between reported dosimetric parameters and increased life span (ILS). In particular, the authors are interested in determining whether the peak dose associated with spatially fractionated RT (SFRT) is the most important parameter related to treatment effect. This is an important question as the current understanding between dosimetry and treatment response in SFRT is very limited and the paper brings forth many good points.  It is well written and organized with no major typographical or grammatical errors. The authors suggest that valley dose is the most important parameters related to ILS, which agrees with several other publications.

I have two major points I would like the authors to consider for revision in addition to several minor points outlined later:

Major point #1: There are at least 3 instances within the manuscript where the authors suggest that SFRT has been shown to be superior to conventional RT in clinical applications:

  • In the abstract, lines 19-21: “Preclinical and clinical studies have shown that SFRT can produce a significantly higher therapeutic index than conventional radiotherapy (RT).”
  • In the introduction, lines 40-42: “Compelling evidence from both preclinical research and clinical studies have shown that SFRT can lead to a very high therapeutic ratio compared with conventional RT”
  • In the Discussion, lines 237-238: “Decades of studies have shown that SFRT can have a significantly higher therapeutic index compared with conventional RT.”

To my knowledge, there have not been any clinical patient treatments with microbeam or minibeam RT so these modalities could not be considered advantageous over conventional RT in the clinical setting. The only actual in-human treatments have used larger scale “conventional” SFRT methods like lattice RT or grid therapy and of these I am unaware of any clinical randomized controlled trials comparing SFRT to conventional RT. Indeed, as the authors later summarize in lines 254-257, reference #5 states “However, there are also important limitations of the studies described. In most cases, interpretation of these studies is confounded by (1) lack of a control arm;…"

It may be misleading then to state that SFRT has been shown to be better than conventional RT, as there is no strong data yet to support this. There certainly may be preclinical data and anecdotal evidence in the literature, but I suggest the authors revise the bulleted statements above. Doing so would not detract at all from their message, and has the advantage of striking a more neutral tone since there is no comparative data in the setting of a randomized controlled trial to support the notion SFRT is superior to conventional RT. Please note I am not disputing the idea that SFRT may be superior in some instances, I am only suggesting there is no data which has directly confirmed it in patients.

  • We thank the review for this thoughtful input.
  • We have addressed the 3 points above by either adding more explanatory sentences or toning down the statements so as not to mislead readers.

Major point #2: I would like the authors to reconsider, or at minimum clarify, the wording and statistical interpretation of their results with regard to correlation and significance of p values.

o    For example in line 197: “Average dose and peak dose also show significant correlation with ILS…”

The Pearson coefficient is an indication of correlation, but the p-value doesn’t really tell you much without additional context related to the strength of that correlation. For example, a highly significant p value doesn’t tell you anything about the strength of the correlation and therefore nothing about its practical importance. In other words, I would argue a significant p value for a low value of correlation is not necessarily clinically relevant, especially for the small sample sizes used in this paper. I suggest the authors reframe the wording of their results and cite what is considered strong/weak/no correlation to aid in their interpretation. Consider Cohen J. Statistical Power Analysis for the Behavioral Sciences. 2nd ed. Hillsdale, NJ: Lawrence Erlbaum Associates; 1988 where a correlation of 0.3 - 0.5 is generally considered weak correlation, 0.5-0.7 is considered moderate correlation, and higher values are considered strong correlation. With regard to MRT and peak dose, this changes the message somewhat because the authors could argue peak dose is only “weakly correlated” despite a significant p value. However, valley dose would be “strongly correlated” with ILS and has a significant p value which further supports the idea that valley dose is most important. Similar arguments could be made for MBRT and valley dose as well.

  • We sincerely thank the review for this suggestion. We have fully incorporated the operational definitions “small” “medium” and “large” in our manuscript.
  • You will find the extra information in Data Analysis and in the Legend of the tables.
  • We have also adapted the manuscript to incorporate the new operational definitions.

 I would ask the authors to please consider the following minor suggestions as well:

  1. Line 89: “To date, all SFRT regimes have been prescribed according to the peak dose…”

Please consider avoiding the term “all”, I would suggest “most” instead. There are lattice SFRT protocols that do not prescribe to the peak dose. See for example, Duriseti, S., et al. (2021). "Spatially fractionated stereotactic body radiation therapy (Lattice) for large tumors." Adv Radiat Oncol 6(3): 100639. There are other lattice SFRT papers which do not prescribe to peak dose as well.

  • It seems there is a misunderstanding here, with ALL, we refer to “all” pre-clinical SFRT parameters selected in this study, we are not referring to “all” the available data nor to the clinical data. We have added a sentence to clarify this in the legend of Table III:

“The group “ALL” includes all pre-clinical SFRT parameters (variables) selected in this study, which are presented in Tables I and II.”

  •  
  1. Lines 143-145: “Peak doses of less than 800 Gy were used in the studies. This criterion was intended to avoid confounding factors related to toxic effects of radiation exposure that are unlikely to be relevant to the clinical application of SFRT.”
  • I found the choice of 800 Gy to be interesting, especially since one might think toxic effects of radiation exposure in humans could result with peak doses much less than that. Could the authors comment on why 800 Gy was chosen as the cutoff? In particular, what toxic effects are of concern that may manifest only above 800 Gy? If this was an arbitrary cutoff, I think that would be completely fine actually.
    • This cutoff was mostly arbitrary (not based on empirical data). We simply wanted to avoid having data that was too spread and affect the correlations.  
  • At times throughout the manuscript, I was somewhat confused as to what modality was being referred to since clinical grid therapy/lattice, MBRT and MRT are all generally characterized as SFRT here. Is there a way to separate these 3 applications of SFRT so it is more clear to the reader which one is being referred to? As an example, consider the statement "...unlikely to be relevant to the clinical application of SFRT." Do the authors mean clinical application of MRT or MBRT? Grid therapy (which is also SFRT) is already in clinical use.
    • Thank you for this. We went over the manuscript and in places where SFRT was mentioned, we made sure that it was accompanied by the words “clinical” (meaning GRID and LATTICE) or “preclinical” (meaning MRT and MBRT) to reflex the explanation that we show in Figure 1. We hope this now facilitates the understanding of the reader.

  1. Lines 159- 160: “We also included a few unpublished studies soon-to-be published to increase statistical power.”
  • Can the authors please define “soon to be published?” Have these articles been accepted for publication but are not in press yet? If not, I don’t think they should be included in the analysis until accepted in a peer reviewed journal.
    • We did 2 things: 1) one of the articles is currently accepted for publication (we updated this in the table), 2) we eliminated the unpublished data that has not been submitted for publication yet.
    • We re-ran all the analysis and updated the tables accordingly. There were no major changes to the results.
  • Table 1: All but one of the “Average Dose (Gy)” values are zero. Is this a mistake? Please clarify.
    • This was a mistake when transferring the data into the manuscript. Those “zeros” had a formula and for some reason the values were not pasted. We have corrected this in the table. The calculations were not affected by this.

  1. Line 200: “…(see Figure S2 in the supplementary material).”

Supplementary materials are not available for review. Will they be provided?

  • We provided the supplementary material during the submission process. We don’t know what it didn’t reach the reviewer. We recommend requesting it to the Editor.

  1. Lines 248 – 250: “We hypothesize that the dosimetric parameters that capture the essence of SFRT dosimetry (i.e., the coexistence of hot and cold dose sub-regions) will have a closer correlation with treatment response…”
  • I agree with this statement, but it is contradictory to lines 203-204: “Our findings challenge the long-held assumption that PVDR plays an important role in the treatment response of MRT [21], which has been widely shared among MRT researchers.” I would suggest most readers would associate “the coexistence of hot and cold dose sub-regions” with PVDR.
  • PVDR is often considered a measure of heterogeneity, and heterogeneity is often considered to be very important for treatment effect. But your analysis suggests there might not be any correlation between ILS and heterogeneity which is quite interesting and perhaps a bit provocative. Would the authors consider a bit more discussion on this potential disparity? Does this mean that PVDR is not a good metric of heterogeneity or that heterogeneity is just not important for the biology?
    • Thank you for your thoughtful analysis. Our results show that PVDR is relevant for the physics of the spatial fractionation but do not seem to be a direct driver of biological effect (ILS). We have added the following 2 sentences in the discussion to expand on this:

“This is demonstrated by our review showing that PVDR, a SFRT-specific dosimetric parameter that characterizes a major aspect of radiation spatial fractionation, has no significant correlation with ILS. Further careful studies are needed to unravel the correlation between SFRT dosimetry and treatment response.”

  • Regarding your question whether heterogenicity is not important for biology, we believe that our results cannot answer that. The reason is that besides ILS, we have no other biological variable that we can measure in this study. This is important because to evaluate the effects of heterogenicity, we would need to have data about for example: immune cell influx, normal tissue effect, tumor cell turnout, etc. Therefore, an whole new experiment should be designed with the sole purpures of studying whether PVDR is a good metric to evaluate the biological effect of heterogeneity.

  1. Line 257: “In particular, dose prescription is usually at the surface…”

I am not aware of any clinical SFRT treatments which prescribe dose at the surface of the patient and this would be very difficult to do as most TPS’s have questionable accuracy right at the skin surface. Can the authors please clarify?

  • This is now clarified in the discussion.

  1. Lines 261-262: “Lattice therapy is a significant improvement compared to GRID therapy, as the 3D dose vertexes (peaks)…”

I suggest the more commonly used term "vertices" rather than vertexes.

  • We have adapted the manuscript as requested

Reviewer 2 Report

This study was reported the valley dose was associated with ILS in patients who received SFRT. Overall, this paper is well written. The reviewer thinks that this report has useful information for readers. The reviewer would like to suggest some critiques as follows.

1.     On line 26, the authors should delete “space.”

2.     On line 32, what is best with treatment outcomes? The authors should revise this point clearly.

3.     On line 75, “the mechanism of action ….. conventional RT.” Is unclear. What is different? The reviewer wants specific information about “very different.”

Author Response

REVIEWER 2:

This study was reported the valley dose was associated with ILS in patients who received SFRT. Overall, this paper is well written. The reviewer thinks that this report has useful information for readers. The reviewer would like to suggest some critiques as follows.

  1. On line 26, the authors should delete “space.”
    1. Done
  2. On line 32, what is best with treatment outcomes? The authors should revise this point clearly. 
    1. Thank you. We have improved the clarity of this sentence.
  3. On line 75, “the mechanism of action ….. conventional RT.” Is unclear. What is different? The reviewer wants specific information about “very different.”
    1. Thank you. We have added two more sentences expanding on this difference

Round 2

Reviewer 1 Report

The manuscript has improved and most of the suggested changes have been implemented. I would like to thank the authors for their careful review and consideration of my comments. I have one remaining issue to be considered, and found a number of minor typos in the revision:

1. Abstract: "While in preclinical studies SFRT can achieve a significantly higher therapeutic index than conventional radiotherapy (RT), studies have shown that SFRT followed by RT, can result in a higher therapeutic ratio than either treatment alone."

I feel this statement could again be somewhat misleading and is a bit confusing to read. Are the authors referring to preclinical or clinical studies when they say "studies have shown that SFRT..."? The sentence suggests there has been a study which directly compared SFRT followed by additional RT to either SFRT alone or RT alone. To my knowledge, there has never been such a clinical study in patients so I would ask the authors to please provide a reference for this, clarify, or reword this statement as was done similarly in the previous revision to avoid confusion. Are the authors referring to a planning study perhaps?

2. Line 78: "Cytotoxic radiation cell killing that is dominate in seamless..." Suggest replacing "dominate" with dominant.

3. Lines 81-82: "these secondary effects may play much significant roles compare to conventional RT." Suggest replacing "much significant roles" with  "much more significant roles compared to conventional RT".

4. Line 188, Figure 1 caption: Suggest replacing "Comparativally" with comparative.

5. Line 189, Figure 1 caption: Suggest replacing "to produced these plots" with "to produce these plots".

6. Line 195: "...come from different of animal models." Suggest deleting "of".

7. Line 236, Figure 2 caption: Suggest replacing "off" with "of".

Author Response

REVIEWER 1

  1. Abstract: "While in preclinical studies SFRT can achieve a significantly higher therapeutic index than conventional radiotherapy (RT),studies have shown that SFRT followed by RT, can result in a higher therapeutic ratio than either treatment alone."

I feel this statement could again be somewhat misleading and is a bit confusing to read. Are the authors referring to preclinical or clinical studies when they say "studies have shown that SFRT..."? The sentence suggests there has been a study which directly compared SFRT followed by additional RT to either SFRT alone or RT alone. To my knowledge, there has never been such a clinical study in patients so I would ask the authors to please provide a reference for this, clarify, or reword this statement as was done similarly in the previous revision to avoid confusion. Are the authors referring to a planning study perhaps?

  • We thank the reviewer for reading our manuscript so thoroughly. We revised the sentence and made it more objective to avoid misleading the reader. We hope this new version satisfies the reviewer.
    • “In preclinical studies using single-fraction treatment, SFRT can achieve a significantly higher therapeutic index than conventional radiotherapy (RT). Published clinical studies of SFRT followed by RT have reported promising results for bulky tumors. Several clinical trials are currently underway to further explore the clinical benefits of SFRT.”
  1. Line 78: "Cytotoxic radiation cell killing that is dominate in seamless..." Suggest replacing "dominate" with dominant.
  • Thank you, we have fixed this typo.
  1. Lines 81-82: "these secondary effects may play much significant roles compare to conventional RT." Suggest replacing "much significant roles" with  "much more significant roles compared to conventional RT".
  • Thank you, we have amended the sentence.
  1. Line 188, Figure 1 caption: Suggest replacing "Comparativally" with comparative.
  • Thank you, we have fixed this typo.
  1. Line 189, Figure 1 caption: Suggest replacing "to produced these plots" with "to produce these plots".
  • Thank you, we have fixed this typo.
  1. Line 195: "...come from different of animal models." Suggest deleting "of".
  • Thank you, we have fixed this typo.
  1. Line 236, Figure 2 caption: Suggest replacing "off" with "of".
  • Thank you, we have fixed this typo.